# Chronic Cavitary Pulmonary Histoplasmosis–Novel Concepts Regarding Pathogenesis

**DOI:** 10.3390/jof11030201

**Published:** 2025-03-05

**Authors:** John F. Fisher, Michael Saccente, George S. Deepe, Natasha M. Savage, Wajih Askar, Jose A. Vazquez

**Affiliations:** 1Division of Infectious Diseases, Medical College of Georgia, Augusta University Medical Center, Augusta, GA 30912, USA; jvazquez@augusta.edu; 2Division of Infectious Diseases, University of Arkansas for Medical Sciences, Little Rock, AR 72205, USA; saccentemichael@uams.edu; 3Division of Infectious Diseases, University of Cincinnati Medical Center, Cincinnati, OH 45229, USA; deepegs@ucmail.uc.edu; 4Department of Pathology, Medical College of Georgia, Augusta University Medical Center, Augusta, GA 30912, USA; nsavage@augusta.edu; 5Division of Infectious Diseases, Banner University Medical Center, University of Arizona, Phoenix, AZ 85006, USA; waskar5@hotmail.com

**Keywords:** cavitary, histoplasmosis, pathogenesis

## Abstract

Because the apices of the lungs are most commonly involved in chronic cavitary histoplasmosis (CCPH), it has been assumed by many to have a pathogenesis which is similar to post-primary tuberculosis. Fungi such as *Aspergillus* may colonize pulmonary bullae. Although less common, colonization by *Histoplasma capsulatum* in a heavily endemic area is possible or even probable. In chronic obstructive pulmonary disease (COPD), apical bullae are characteristic. Since COPD is common and CCPH is rare, the pathogenesis of CCPH remains incompletely understood. What is presently known about the pathogenesis of CCPH has not changed appreciably since 1976. A cellblock from a patient with CCPH was analyzed with histochemical stains for T cells, B cells, plasma cells, and macrophages to better understand the pathogenesis of CCPH. The pathogenesis of cavitary disease in histoplasmosis has been assumed to resemble that of tuberculosis. However, liquefaction of a caseous focus in lung apices which resulted from blood-borne tubercle bacilli is distinctly unlike CCPH, as caseation is unusual. Rather, repeated colonization of the apical and other bullae by propagules (microconidium, macroconidium, hyphal fragment) of *H. capsulatum* in patients with COPD who have resided in heavily endemic areas appears to be the primary event in CCPH. Immunohistochemical enumeration of specific cell types in a patient with CCPH has not been previously carried out to our knowledge, but is only a first step in understanding the disease. In future studies, identification of the varieties of macrophages and cytokines in CCPH may reveal whether the process is pro-inflammatory, anti-inflammatory, or both.

## 1. Lay Summary

The fungus *Histoplasma capsulatum* (*Histoplasma capsulatum* is used throughout the article in lieu of current species taxonomy, *H. ohiense*, *H. mississippiense*, *H. suramericanum*) causes a lung infection, common in the midwestern U.S. but, in most persons who reside in these regions, there are usually no symptoms and long-term immunity develops. A few patients with emphysema and COPD uniquely develop cavities. The authors explain why.

## 2. Introduction

Pulmonary infection due to fungi is common in areas of the world whose soil or caves are heavily colonized by the thermally dimorphic fungus, *Histoplasma capsulatum* [1]. The vast majority of infections are asymptomatic or never recognized because the constellation of symptoms mimics a flu-like, lower respiratory tract illness that spontaneously resolves [2]. Clinical syndromes vary according to minimal, moderate, or heavy exposure to conidia or hyphal fragments of the fungus and are diagnosed by microscopy of body fluids, tissue samples, cultures of respiratory secretions, urine or serum antigen detection, or the demonstration of high-titer antibodies to various fungal components [2]. Long-term residence in endemic areas is very likely to result in exposure at some point, with demonstrable serum antibodies in most persons with normal innate and adaptive immunity, whether or not clinical disease ever occurred [1,2,3].

Some long-term residents of these areas develop various forms of chronic lung disease in parallel with frequent exposure to the propagules (microconidium, macroconidium, hyphal fragment) of *H. capsulatum*. Particularly notable among these residents are those with chronic obstructive pulmonary disease (COPD) who manifest structural abnormalities, such as bullae or bronchiectasis of the lung. Clearance mechanisms in such lesions are generally abnormal. Moreover, some patients with COPD develop a unique form of pulmonary infection known as chronic cavitary pulmonary histoplasmosis (CCPH) [4]. This disease is characterized by pulmonary and constitutional symptoms similar to tuberculosis, but to a lesser degree [4,5]. Furthermore, because the apices of the lungs are most commonly involved in CCPH, it has been assumed by many to have a pathogenesis which is similar to that of post-primary tuberculosis. However, it is well known that cavitary tuberculosis is preceded by blood-borne organisms which seed the apices of normal or diseased lungs. There, tubercle bacilli find a safe harbor to later proliferate and ultimately produce apical fibronodular lesions or cavities. Although *Histoplasma* yeast can disseminate in the blood in both normal and immunocompromised persons, in the former, resolution is usual and calcified granulomas can be found in many organs. The infection is ordinarily progressive in the latter group unless treated. Solid evidence of cavitary breakdown with caseation from quiescent foci in the apices is lacking. In addition, there appears to be no equivalent counterpart to CCPH in pulmonary coccidioidomycosis.

It is our intent to clarify some of the unanswered questions regarding the pathogenesis of CCPH, contrasting it where appropriate with both pulmonary tuberculosis and coccidioidomycosis. To that end, we have posed the following questions:What is the nature of the inflammatory response in the pulmonary cavities of patients with CCPH employing immunohistochemical staining for cell types?Do previous bullae become the so-called “marching cavities” of CCPH after new, repeated, or continuous exposure to *H. capsulatum*?What is the explanation for the greater yield from fungal cultures of *H. capsulatum* from patients with CCPH as compared to the reported yield from other forms of pulmonary histoplasmosis?Do patients with chronic pulmonary histoplasmosis but without COPD develop cavities during pulmonary histoplasmosis?Are the inflammatory properties of the lung microbiome actually responsible for much of the cavitary disease previously attributed to CCPH in endemic areas?How does the pathology of CCPH differ from that of cavitary tuberculosis?Do patients with COPD develop cavitary disease resembling CCPH, but caused by *Coccidioides* spp., when residing for long periods of time in the heavily-endemic, desert Southwest?

We attempted to address the above questions by a review of the literature and with lung tissue acquired from a patient with CCPH, employing both histopathology and immunohistochemical analysis of the tissue section.

## 3. Materials and Methods

A cellblock of lung tissue from an autopsy of a patient known to have had CCPH was located in the archives of the Department of Pathology at the University of Arkansas for Medical Sciences. The cellblock was de-identified and cut at 7µ, typically one cell thick, one section per slide, and one slide per cell block. Sections were then stained as follows: hematoxylin and eosin, tissue Gram stain, Gomori methenamine silver, acid-fast, CD20 (B-cells), CD3 (T-cells), CD138 (plasma cells), and CD163 (macrophages).

Histopathology and immunohistochemical findings were analyzed and described by one of the authors (NMS).

## 4. Results and Discussion


**1. What is the nature of the inflammatory response in the pulmonary cavities of patients with CCPH employing immunohistochemical staining for cell types?**


There was a diffuse neutrophilic, acute inflammatory response with associated necrosis. However, the tissue was also replete with macrophages, T lymphocytes, and budding yeast (Figure 1). Most of the B lymphocytes were perivascular. Plasma cells were rare and in no particular pattern. Occasional, scattered Gram-positive bacilli were seen in the specimen, but acid-fast stains were negative. No caseation or hyphal organisms were present in the section. While the specimen from our patient revealed a cavity with a thickened, fibrotic capsule, it could not be determined whether the capsule was a remnant from a former bulla or had developed in previously normal lung.

A perivascular of B cells has been noted in periventricular blood vessels of patients with multiple sclerosis, indicating that B cells are essential contributors to immune responses involved in that disorder [6], but the significance of this finding in our patient is unknown. It is known that B cells and mediators which activate them are found in increased numbers in COPD, especially in the emphysema phenotype, but they may also have a protective role in the setting of acute exacerbations. Their contribution to CCPH in our patient is unknown.

We did not distinguish whether macrophages were of the M1 or M2 type, which might have indicated the predominant pro-inflammatory or anti-inflammatory nature of the inflammation present in our specimen [7].


**2. Do previous bullae become the so-called “marching cavities” of CCPH after new, repeated, or continuous exposure to *H. capsulatum*?**


Goodwin and colleagues provided a painstakingly complete histologic and radiographic description of the clinical pathogenesis of CCPH in 1976, which has not been further expanded upon since, and immunohistochemical research was in its infancy at the time [4]. Nevertheless, more recent updates continue to cite their descriptive data. The authors divided their observations into early findings, and late, chronic disease.

Especially common in early disease was the presence of the interstitial inflammatory process near apical bullae, the walls of which were often thickened. Conspicuously near to the bullae, there were also scattered focal or isolated areas of infarct-like necrosis from occlusion or compression of small and medium-sized arteries and arterioles. Giant cells and mature granulomas were not characteristic of early disease. The majority (80%) of early lesions did not cavitate and healed completely. However, there was an apparent interaction between a pre-existent bulla and interstitial pneumonitis. Both are open to ambient air [8]. This suggests either that the original antigenic stimulus was present in pulmonary parenchyma and involved an adjacent bulla, or the reverse. The abundance of yeast in bullae and paucity in the interstitium is more consistent with the bulla being the site of initial infection, yeast proliferating, and the ensuing inflammation spreading to adjacent parenchyma.

The remaining 20% developed persistent cavitation and slowly progressive, chronic disease instead of a self-limiting process if the early pneumonitis occurred near bullous spaces. The thickness of cavity walls was typically 2 to 3 mm, the inner lining of which was necrotic and the outer wall fibrotic. Yeast were found only in the necrotic lining or in the surface exudate, often within mononuclear phagocytes, suggesting that more abundant antigen produced a response which led to a corresponding increase in wall thickness. Giant cells and mature tubercles were more common in late disease but rarely to the degree found in tuberculosis.

In cavities showing the thickest walls radiographically, continuing necrosis of the wall was evident pathologically, and progressively enlarged. To the authors, this process appeared to “march” through the adjacent lung as the advancing margin of apical cavities destroyed contiguous lung tissue, and “spill over” into adjacent, dependent areas of the lung. Thus, an important part of the process is that the larger apical cavities affect dependent areas of the lung, because the spillover is likely to contain large quantities of antigens. Adjacent fibrosis was exuberant and reminiscent of the mediastinal fibrosis in some patients, with the superior vena cava syndrome associated with histoplasmosis [9].

Such observations suggest the intriguing possibility that the entire process may have begun in an area of emphysema open to ambient air and colonized by a propagule (microconidium, macroconidium, hyphal fragment) of *H. capsulatum*, especially in a heavily endemic area [8]. COPD patients who reside in these locations are likely to have developed a vigorous innate and adaptive immunity in the remote past, which is frequently re-challenged [3]. That challenge may be magnified in many of these home-bound patients, who must remain inside, but are potentially exposed to high numbers of conidia from air conditioning units [10]. The close association of CCPH with COPD suggests that, as lungs developed bullae in the process of progressive COPD, the patients became predisposed to the formation of CCPH. Continued residence in endemic areas is likely to result in repeated infections of normal lung, as well as in pulmonary bullae. When it is considered that 80% of patients with COPD have bullous emphysema, the likelihood of an association with CCPH seems less remote [11,12]. To date, there are no experimental data to support such a hypothesis and no suitable animal model to test it. Moreover, the formation of cavities in other patients with histoplasmosis cannot be explained by a similar process at present.

In endemic regions where re-infection is likely, colonization of such bullae by *H. capsulatum* can potentially occur [3] and would even be likely. It is therefore probable that inflammation in and around a bulla would follow and superficially resemble cavitary tuberculosis. However, in post-primary tuberculosis, apical lung cavities are the result of the caseous process in persons with intact cell-mediated immunity [13,14,15,16]. In CCPH, apical bullae were already present because of bullous lung disease in patients with COPD. These bullae were likely to have been re-infected by the inhalation of *H. capsulatum* propagules in endemic areas. Moreover, the immune system of most of these patients is likely to have been continually re-challenged over prolonged periods [3]. Therefore, parenchymal pneumonitis surrounding a previously existing bulla or bleb colonized by *H. capsulatum*, developing into a thin- or thick-walled cavity, would be a more plausible pathogenesis. Thus, it appears that some cavities were once bullae or large emphysematous spaces, which became infected.

Regardless of origin, the tendency for colonizing infection of old pulmonary cavities with *Aspergillus* spp. and the development of “fungus balls” (aspergillomas) is well known and readily recognized radiographically. The risk of acquiring an aspergilloma in a cavity >2 cm in diameter is 15–20% [17]. In bullous emphysema, many bullae of this size are characteristic and, since they are open to the airway, it is not surprising that colonizing aspergillosis has occurred in this disease [18,19]. Secondary aspergillus infection with cavity wall thickening without a fungus ball in healed or apparently-healed cavities of histoplasmosis are less well known, but have been reported and are most common in the Tennessee area [4].

Such colonization is likely to have been related to the density of airborne conidia of *Aspergillus* in the area. The usual airborne conidia counts of *A. fumigatus* are 200–500 conidia/cubic meter, reflecting counts in soil, which range from 9.5 × 10^4^ colony forming units (CFU)/g to 5.5 × 10^5^ CFU/g [20]. In contrast, in heavily infested soil, the number of propagules of *H. capsulatum* has been estimated to reach 10^5^/g of soil [21,22]. As expected, the usual airborne counts of *H. capsulatum* are much lower. Over the course of six days, only 19 airborne conidia from an endemic area were recovered by Rooks in 1954 from a rooftop device in Iowa City [23], the maximum rate of deposition being six per hour. At each hour’s reading, approximately 1 cu. yd. of air was sampled. This result, if the dosage were maintained over a 24-h period, would mean an inhalation rate of *H. capsulatum* conidia, for the adult at rest, of approximately 100 during normal activity. Since only tuberculate macroconidia were counted, the actual number of conidia was likely to have been much higher [3]. In light of these data, a colonizing form of histoplasmosis could theoretically explain the pathogenesis of CCPH in some patients with COPD.


**3. What is the explanation for the greater yield from fungal cultures of *H. capsulatum* from patients with CCPH as compared to the reported yield from other forms of pulmonary histoplasmosis?**


In mild to moderate acute pulmonary histoplasmosis, *H. capsulatum* is uncommonly isolated from sputum [3]. However, in CCPH patients with thin-walled cavities, sputum cultures were positive in 35% and, in those with thick-walled cavities, 57% [4]. These results were consistent with findings in a later series of patients with CCPH, in which various specimens yielded the organism in 57.5% [24]. The higher yield of cultures in CCPH is unexplained and could be due to poor pulmonary clearance mechanisms, an incompletely effective adaptive immune response during repeated exposures to the fungus, or a colonizing form of infection, like that encountered in pulmonary aspergillosis.

Fungi such as *Aspergillus* spp. are notorious for colonizing lung cavities with culture positivity in at least one-fourth of patients [25]. Whether *H. capsulatum*, like *Aspergillus* spp., may colonize and proliferate in cavities in heavily endemic areas, explaining the greater frequency of positive cultures than other forms of pulmonary histoplasmosis, is purely speculative.


**4. Do patients with chronic pulmonary histoplasmosis but without COPD develop cavities during pulmonary histoplasmosis?**


While cavitary disease in pulmonary histoplasmosis occurs overwhelmingly in patients with COPD or in patients with a heavy exposure to tobacco smoke, cavities can certainly be part of the pulmonary manifestations of persons without COPD and in those who have never smoked [26], including some pediatric patients [24,27]. No details were available from these patients as to the presence of underlying lung disease. However, such patients are less likely to exhibit pulmonary cavities according to Kennedy and Limper. Nevertheless, pulmonary cavities were present in 9 of their 30 patients who underwent computed tomography (CT) [26].

Presumably, cavities developed from *H. capsulatum* infection in previously normal lung tissue and not by a process similar to that of patients with COPD. At the same time, it should be noted that the prevalence of bullae in an otherwise healthy population up to 40 years-of-age is 7.2% [28]. In heavily endemic areas, the colonization of such bullae by *H. capsulatum* could theoretically lead to progressive inflammation and ultimately to a cavity, as in patients with COPD, but on a far smaller scale.


**5. Are the inflammatory properties of the lung in COPD, as well as the microbiota, actually responsible for much of the cavitary disease previously attributed to CCPH in endemic areas?**


It has been suggested that, in COPD, there is an age-associated decline in immunity, which is important in the development of the notorious structural abnormalities which are present [29,30]. In many older persons, innate phagocytes, such as neutrophils, macrophages, dendritic cells, and natural killer cells, show a measurable decline in phagocytosis and chemotaxis. In addition, their ability to secrete pro-inflammatory cytokines and present antigens is diminished. There is also a gradual switch from a Th1 immune response to that of a Th2 profile with exposure to immunostimulants, leading to an overproduction of Th2 cytokines. A Th2-predominant profile can also lead to a low-grade, chronic, systemic inflammation known as “inflamm-aging” provoked by a continuous antigenic load and stress. Aging of the immune system can directly affect macrophage activation. Aging macrophages exhibit a functional decline in phagocytosis and chemotaxis and in their ability to secrete pro-inflammatory cytokines and kill microorganisms which are part of the microbiota of the respiratory tract. Residents of endemic areas who also suffer from COPD are likely to inhale propagules of *H. capsulatum*, at least intermittently [3]. Theoretically, the response of aging macrophages to fungi should also be diminished. In addition, the barrier function of the respiratory epithelium wanes in the elderly. The aggregate decreased ability to protect the lungs from inhaled particles and microbes can lead ironically to more inflammation. Such a decline is believed to be mostly responsible for the presence of a subclinical, chronic inflammatory process in elderly patients with COPD.

Whether the plentiful, visible yeast noted in our patient’s specimen is partially explained by the response of aging macrophages is speculative. However, the observation is consistent with the frequency of positive cultures in CCPH. Obviously, the role of alveolar macrophages, dendritic cells, other phagocytes, and cytokines in the lung tissue of persons with CCPH, who are repeatedly exposed to *H. capsulatum* in an “inflamm-aging” cavity, would be of interest in understanding the pathogenesis of the entire process.

Normal, as well as diseased airways are not sterile, as was previously believed [31,32,33]. Unregulated inflammation is one of the factors responsible for many chronic lung diseases, including COPD. Certain taxonomic groups of pro-inflammatory bacteria, such as *Klebsiella pneumoniae* and *Pseudomonas aeruginosa*, are often part of the microbiota of these persons; on occasion, even *Pneumocystis jirovecii* has been isolated [32,33,34]. In addition, there have been reports of a significant decrease in bacterial diversity when the presence of *P. aeruginosa* has been detected [35,36]. Such pulmonary dysbiosis could provide the constant inflammatory stimulus that has long been observed in COPD. Conceivably, in an endemic area, intermittent exposure to *H. capsulatum* could compound the dysbiosis. On the other hand, progression of cavitary disease in histoplasmosis, even in culture-positive persons, could be falsely attributed to the fungus and be more related to other members of the airway microbiota.

Scattered, rare Gram-positive bacilli were present in our tissue samples. The additional contribution of these organisms to the inflammatory process is unknown. At a minimum, the presence of these organisms is consistent with communication of the cavity with the outside airway [8]. However, given the abundance of yeast present, most of the inflammation was more likely to have been a response to the presence of *H. capsulatum* and not to bacteria.


**6. How does the pathology of CCPH differ from that of cavitary tuberculosis?**


While caseation has been described, it must be the exception in any form of pulmonary histoplasmosis, because caseation necrosis in tuberculosis has a unique pathogenesis. In post-primary tuberculosis, caseation follows the development of an obstructive, lipid pneumonia in primarily small, apical pulmonary lobules, which results from the interaction of 6, 6’trehalose di-mycolate (cord factor) on the surface of *Mycobacterium tuberculosis* with surfactant. If the obstruction involves a large enough area of a pulmonary segment, caseation necrosis liquefies, forming a cavity [13,14,15,16]. Granulomas and giant cells are abundant. No such caseating process has been described in histoplasmosis.

The entire pulmonary section from our patient revealed a cavity with a fibrotic capsule and adjacent necrosis. There was little adjacent normal pulmonary tissue. No caseation necrosis and no granulomas were present in the specimen. The development of a cavity in adjacent normal lung remains a possibility. However, the pathological process in our patient’s specimen is distinctly unlike the pattern which ordinarily occurs in tuberculosis.


**7. Do patients with COPD develop cavitary disease resembling CCPH, but caused by *Coccidioides* spp. when residing for long periods of time in the heavily endemic desert Southwest?**


The prevalence of COPD in the lower Sonoran life zone of 4.7–5.4% is lower than that of residents of the Ohio and Mississippi River valleys at 6.6–11.9% (Data Source: CDC Behavioral Risk Factor Surveillance System (BRFSS), 2020). However, with respect to the incidence of acute pulmonary coccidioidomycosis, in 2019, there were 20,003 cases of Valley fever reported to the CDC. Cases rose from 84.4 cases per 100,000 population to 144.1 per 100,000 from 2014–2019.

In 2019 also, there were only 1124 cases of confirmed or probable acute pulmonary histoplasmosis. The incidence of acute histoplasmosis is estimated to be 6.1 cases per 100,000 (Data Source: CDC Behavioral Risk Factor Surveillance System (BRFSS), 2020). In a series of 21 patients with cavitary, pulmonary coccidioidomycosis who underwent surgical resection of a coccidioidal cavity, 11 patients (52%) had a smoking history, but COPD was not listed among the comorbidities of their patients [37]. Unlike CCPH, cavity walls showed neutrophils and caseous necrosis in 20 of 21 cases. Hyphal forms of *Coccidioides* spp. were found in 62%. Similarly, spherules were present in 76%. In contrast, the presence of hyphal forms of *H. capsulatum* has been reported in endovascular infections [38], but not to our knowledge in CCPH. Further, unlike CCPH, positive cultures were the exception in cavitary coccidioidomycosis, with only a single isolate among the 21 patients [37]. A cavity in coccidioidomycosis with a fungus ball/mycetoma was not a rarity, as in CCPH, occurring in 6 of 21 patients.

Thus, despite the increased incidence of acute coccidioidomycosis compared to that of histoplasmosis, a clinical counterpart to CCPH in areas rampant with arthroconidia of *C. posadasii* or *C. immitis* is conspicuously absent.

## 5. Conclusions

The pathogenesis of cavitary disease in histoplasmosis has been assumed to resemble that of tuberculosis. However, liquefaction of a caseous focus in lung apices, which resulted from blood-borne foci of tubercle bacilli, is distinctly unlike CCPH, as caseation is unusual. Rather, repeated colonization of the apical and other bullae by propagules of *H. capsulatum* in patients with COPD who have resided in heavily endemic areas appears to be the primary event in CCPH. The amount of yeast which is present, the participation of other members of the microbiota, and other factors, such as inflamm-aging, are most likely to dictate the level of necrosis and thickness in the wall. Those infected bullae with the thickest walls progress to cavities and seemingly “march” through apical lung regions.

Pulmonary cavities due to acute and chronic *Histoplasma* infection can develop within relatively normal segments of lung, but cavity formation is much more common in the presence of structural lung disease. Bullae are very common in COPD but are present only in a small percentage of normal individuals. The numerous bullae present in COPD predispose to CCPH in rather dramatic fashion, but whether incidental bullae in normal individuals predispose to cavities in histoplasmosis is unknown.

Infected cavities commonly contain abundant yeast which may at least partially explain the frequency of positive cultures compared to acute pulmonary histoplasmosis, since developing cavities communicate with the bronchial tree.

Our immunohistochemical enumeration of specific cell types in a patient with CCPH has not been previously carried out to our knowledge, but is only a first step in understanding the disease. Identification of the varieties of macrophages and cytokines and determination of the presence of anti-histoplasma antibodies is likely to reveal whether the process is primarily pro-inflammatory or anti-inflammatory or a combination. While we were able to assess the cellular response present in our patient’s tissue sample, we did not analyze the immunoglobulin or cytokine response in this pilot study. Additional quantitative and qualitative analysis of the cellblock remains feasible and is warranted.

## Figures and Tables

**Figure 1 jof-11-00201-f001:**
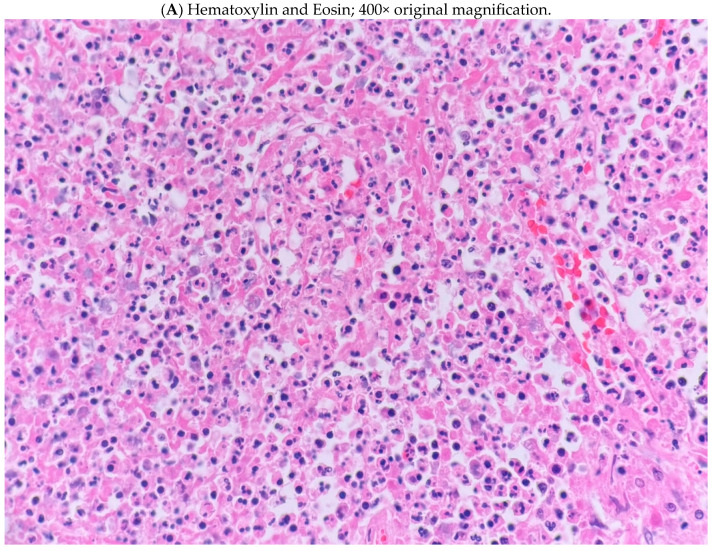
(**A**) Microscopic examination of the tissue section revealed pulmonary parenchyma with a mixed inflammatory infiltrate predominated by macrophages and neutrophils, with associated necrotic debris. Chronic inflammatory cells were also present, including several lymphocytes and a few plasma cells. (**B**) Grocott–Gömöri’s methenamine silver stain; Special staining revealed numerous yeasts, including budding forms. (**C**) Immunohistochemical analysis confirmed the presence of numerous macrophages with abundant cytoplasm (CD163 immunohistochemical stain. (**D**) Although T-cells predominated over B-cells, B-cells were present in a mostly perivascular pattern as noted by immunohistochemical analysis (CD20 immunohistochemical stain). Scale bar: 10 µm.

## Data Availability

The original contributions presented in this study are included in the article. Further inquiries can be directed to the corresponding author.

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
