# Peer review of "Chronic Cavitary Pulmonary Histoplasmosis–Novel Concepts Regarding Pathogenesis"

_jof, 2025, doi:10.3390/jof11030201_

Round 1
Reviewer 1 Report
This manuscript is a detailed review of what is known and conjectured about the pathogenesis of chronic cavitary pulmonary histoplasmosis. Obviously well researched and thoughtfully written. However, the length of the manuscript detracts from the message. Specific comments about reducing the length are noted below.
The Methods note that there were detailed studies performed on tissue from one patient, but no figures are shown to convince the reader about the findings. The Results are mixed with the Discussion; this makes it difficult to evaluate the findings.
1. Introduction: This is very long and is not just an Introduction, but provides details about prior published data by Goodwin et al. and also introduces material that should be placed in the Discussion.
Lines 57-70: Comparing cavitary histo with cavitary tuberculosis should be part of the Discussion and not in the Introduction.
Lines 82-89: Subject matter related to percentage of positive sputum cultures in various infections is repeated and should not be in an Introduction. Perhaps could be deleted?
Lines 116-188: Detailed discussion of work by Goodwin et al could be referred to briefly but not in such detail.
Lines Lines 201-222: Comparing Aspergillus to Histo and discussing spore or conidia counts could be in the Discussion but not the Introduction. Alternatively, this could just be deleted; it seems to be a separate tangent and question if relevant? Paragraph on histo uses the term "spores", but should be "conidia".
2. The authors list 6 questions, but then only discussed 5 of these questions. Question 5, which is not addressed, may not be that relevant and then this subject could be deleted from the Introduction, as well, perhaps.
3. Materials and Methods: These seem truncated in regard to the source of the reagents for staining for specific cell types. Interesting that only one slice was evaluated per cell block-is that adequate?
4. Results and Discussion: These should be separated. Results should be clearly identified and figures provided so the reader can evaluate the results. The Discussion should follow after all the Results are shown. The authors could weave in material from the Introduction into the Discussion, but should try to reference prior literature but not detail so many of the specifics of prior studies.
Lines 338-340: This is not a complete sentence
Lines 357-379: Not sure this is relevant, and I would suggest deleting material about cocci.
Reviewer 2 Report
Fisher and colleagues submit a review regarding pathogenesis concepts for chronic cavitary histoplasmosis, based on detailed review of a single patient’s histopathology. There are no photographs of the histopathology presented as figures, which is quite unexpected.
Comments:
- This reviewer was expecting many figures to illustrate all the histopathology points in the results section. Where are they?
- It would be helpful to explain right in the introduction why patients with CCPH develop both thick & thin walled pulmonary cavities. Most clinicians see certain infections as having one type of cavity (thick or thin, and there were even boards questions related to this issue for a number of years), while other infections had the opposite. See PMID, 18400799, “Cavitary pulmonary disease”.
- It would be helpful to explain in the introduction what a propagule is.
- Isn’t the higher yield of cultures in CCPH due to higher organism burden? The higher organism burden is then what is unexplained?
- Is one reason that your index patient had chronic histoplasmosis rather than disseminated histoplasmosis because he had the presence of T lymphocytes in his tissue? See PMID 6227237, “T lymphocyte abnormalities in disseminated histoplasmosis”.
- Although “Hemoptysis in the absence of chronic bronchitis is rare”, it can also occur with endobronchial histoplasmosis, see PMID 12000819, “Case 14-2002. A 51-year-old woman with recurrent hemoptysis”.
- For “heavily endemic areas”, don’t forget that indoor exposure from air conditioning could be playing a role, see PMID 15983912, “Recurrent exposure to Histoplasma capsulatum in modern air-conditioned buildings”.
- For “Do patients with COPD develop cavitary disease resembling CCPH, but caused by Coccidioides spp…”, Galgiani and Kauffman have a section that contrasts histo & cocci. Cocci seems to stay thin-walled. Is this your experience? See PMID 38324487, Coccidioidomycosis and Histoplasmosis in Immunocompetent Persons.
- I was hoping that the authors would have some speculation on who should receive fungal drug prophylaxis, based on their understanding of the pathogenesis, similar to how models for Aspergillus prophylaxis have been developed.
- Typo: shouldn’t “notorious for colonizing colonize lung cavities” be “notorious for colonizing lung cavities”?
- Administrative point: some spacing and tab issues need to be addressed by administrative staff who are helping with final formatting of the next version of the manuscript.
- Administrative point: should the sentences with “Data Source” at the end actually be referenced in the bibliography?
Round 2
Reviewer 1 Report
This revised manuscript has been shortened and is more "readable"; especially noteworthy is that the Introduction has been drastically cut, which is appropriate. The authors have kept the same format of mixing Results and Discussion, but the revision is easier to follow and does get the intended message across to the reader better than in the original manuscript. Much improved overall. The figures certainly add to the manuscript.
A few comments:
1. I would suggest that the question related to cocci is not relevant to helping to define the pathogenesis of chronic cavitary pulmonary histoplasmosis. I agree both organisms live in soil and cause disease by inhalation of conidia, but this is also true of paracocci and blasto, both of which can cause chronic pulmonary disease. However, no reference is made to those diseases. They really are all different organisms, who just happen to reside in the soil, and I am not sure this question specifically related to cocci is germane to helping to define the pathogenesis of chronic cavitary pulmonary histoplasmosis.
2. Line 57: I think the authors are in error here. They comment that in immunosuppressed persons H .capsulatum can disseminate in the bloodstream and that the infection is progressive unless treated. In fact, H. capsulatum commonly disseminates in the bloodstream to the organs composing the RES in normal hosts as well as in immunocompromised hosts. The latter often cannot handle this and develop disease, but normal hosts handle the infection well when T cell immunity develops and are able contain the organism. Splenic calcifications seen years later are one manifestation of this dissemination to the RES. The work by Goodwin and co-workers and by Jan Schwarz decades ago established this. Would suggest modifying that sentence.
3. Line 111: Word must be missing. Also, not sure that reference to MS is relevant. Would suggest deleting this.
Reviewer 2 Report
Fisher and colleagues present a revised manuscript of chronic cavitary pulmonary histoplasmosis, novel concepts of the pathogenesis.
Comments:
- The figure that was added to the manuscript was unavailable for review.
- Regarding your reply to Comment #7: “For heavily endemic areas, don’t forget that indoor exposure from air conditioning could be playing a role, see PMID 15983912, Recurrent exposure to Histoplasma capsulatum in modern air-conditioned buildings. Reply: We missed this publication in our review of the endemicity of H. capsulatum and could have added it. It would also have been relevant in our comparison of CCPH as we sought a counterpart in coccidioidomycosis where air conditioners are standard for dwellings and businesses.” Why wouldn’t you address this issue in your paper rather than just state that you could have added it? COPD patients get more and more medically debilitated with time, become oxygen-dependent, and sit indoors for years towards the end of their lives?
- Regarding your reply to Comment #8: “Based on our review of Goodwin and colleagues classic paper in 1976, thin-walled cavities of histoplasmosis tend to resolve, while thick-walled cavities progress. So, most of the apical cavities expand and spill antigenic contents into dependent areas of the lung.” Why not add this gem of two sentences to your paper?
- Regarding your reply to Comment #8: “Our manuscript was written before their nice review in the recent NEJM was available.” Does this mean you are unwilling to consider this manuscript in this revision, which is being written a year later than PMID 38324487 was published?
The figure that was added to the manuscript was unavailable for review.
Round 3
Reviewer 2 Report
The authors have addressed my comments with their revisions.
None noted.